# Study on Preparation and Properties of Glass Fibre Fabric Reinforced Polyphenylene Sulphide Composites

**DOI:** 10.3390/ma15249036

**Published:** 2022-12-17

**Authors:** Lingda Shao, Jinbo Huang, Xuhuang Feng, Zeyu Sun, Yingjie Qiu, Wei Tian, Chengyan Zhu

**Affiliations:** 1College of Textile Science and Engineering, Zhejiang Sci-Tech University, Hangzhou 310018, China; 2Zhejiang Sci-Tech University Huzhou Research Institute Co., Ltd., Huzhou 313000, China

**Keywords:** glass fibre, polyphenylene sulphide, composite material, silane coupling agent, mechanical properties

## Abstract

In this paper, glass fiber fabric reinforced polyphenylene sulfide composites were prepared by hot pressing. The effects of glass fibre modification and hot pressing temperature on the properties of the composites were investigated using a scanning electron microscope, infrared spectrometer, universal testing machine, and DIGEYE digital imaging colour measurement system. The results show that after the treatment with a silane coupling agent, the silane coupling agent was more uniformly distributed on the surface of the glass fibres, and the bonding effect between the glass fibre fabric and polyphenylene sulphide was significantly improved. The strength of the composites increased and then decreased with the increase of hot pressing temperature, and the surface colour of the composites became darker and darker. When the hot-pressing temperature is 310 °C, the mechanical properties of glass fabric-reinforced polyphenylene sulfide composites are at their best, the tensile strength reaches 51.9 MPa, and the bending strength reaches 78 MPa.

## 1. Introduction

Polyaryl sulfide is a polymer compound alternately connected by sulfur atom and aryl structure. Polyphenylene sulfide is the most important and widely used polyaromatic sulfide resin [1,2]. Polyphenylene sulfide is a type of special engineering plastic (high-performance engineering plastics) that has the best performance among the plastics with long-term use temperatures ≥150 °C [3,4]. The production and demand of polyphenylene sulfide are second only to polyamide, polyformaldehyde, polycarbonate, modified polyphenylene ether, and polybutylene terephthalate. Polyphenylene sulfide is the sixth largest engineering plastic in terms of usage [5,6,7,8,9].

The molecular structure of polyphenylene sulphide is relatively simple, and the primary molecular chain is alternately arranged by benzene ring and sulfur atoms. Among them, the benzene ring structure of polyphenylene sulphide is rigid, and the thioether bond provides flexibility for polyphenylene sulphide [10,11,12,13]. The appearance of polyphenylene sulphide is white or beige, with high crystallinity, and a hard and brittle polymer [14,15]. The density of pure polyphenylene sulphide is about 1.3 g/cm^3^, and the density will increase after modification. The flame retardancy of polyphenylene sulphide is good, and its oxygen index is as high as 46–53% [16,17,18]. Although polyphenylene sulphide has a high melting point, it has low melting viscosity and good fluidity [19,20,21]. It can be processed by general thermoplastic resin and made into fibres and fabrics, films, coatings, and high-precision products. Since the advent of polyphenylene sulphide, it has been favoured by high-end users in the fields of electronics/electrical, automotive, precision instruments/instruments/machinery, the petroleum and chemical industries, food and textile machinery, national defence, and cutting-edge technology [22,23,24,25,26]. It has become one of the preferred plastic structural materials required in these important industries under extreme conditions such as high temperature, high humidity, strong corrosion, and strong radiation [27,28,29,30,31,32].

The brittleness of polyphenylene sulphide significantly affects its scope of application. Scholars have studied how to improve the brittleness of polyphenylene sulphide [33,34]. There are two main methods of doing so. One is to add the appropriate amount of polymer materials to polyphenylene sulphide for copolymerization modification, such as the polyphenylene sulphide/polyethene terephthalate alloy, polyphenylene sulphide/nylon 6 alloy, polyphenylene sulphide/polyamide alloy, etc. Chen, Chen-Ming et al. synthesized epoxy-modified polyethene (PE-g-GMA) by an extrusion grafting reaction. The mechanical properties of the composites can be effectively improved by adding PE-g-GMA to a polyphenylene sulphide alloy [35]. The other is to fill fibre reinforcement in polyphenylene sulphide, such as glass, carbon, basalt and aramid. Manjunath and Attel prepared E-glass fibre reinforced polyphenylene sulphide composites by compression moulding. The results showed that the polyphenylene sulphide composites filled with 30 wt% glass fibre have the best mechanical properties, such as tensile, bending, impact strength, and dielectric strength [27]. These two methods make up for the deficiency of the performance of polyphenylene sulphide. 

Currently, the cost of copolymerization modification is high, the technology is complicated, and it is challenging to implement in industry. Adding woven fabric reinforcement to polyphenylene sulphide has become an effective way to solve the brittleness of polyphenylene sulphide [36,37,38]. Zhang and Shouyu treated it with plasma to prepare glass fibre reinforced polyphenylene sulphide composites, which improved the mechanical properties of the composites. However, this is a physical method which will cause some damage to the material itself [39]. Jeong, Yeon studied the properties of glass fibre reinforced polyphenylene sulphide composites at different processing temperatures. However, the processing method is screw extrusion, which is unsuitable for preparing large-scale composites [20]. Wang, Wenchao studied the thermal ageing properties of glass fibre reinforced polyphenylene sulphide composites at 250 °C, but the temperature was relatively straightforward [37].

There are few studies on the effect of processing temperature on the properties of laminated composites, and much attention has been paid to the modification methods which do not affect the basic properties of the materials. In this paper, glass fibre fabric was used as the reinforcement and polyphenylene sulphide as the matrix, glass fibre fabric was treated with silane coupling agent KH560, and composites were prepared by the laminated hot pressing method. The effect of process parameters on the properties of glass fibre fabric-reinforced polyphenylene sulphide composites was studied.

## 2. Experiment

### 2.1. Materials

Polyphenylene sulphide particles with a density of 1.35 g/cm^3^ were provided by Zhejiang Xinhecheng Co., Ltd. (model 11100F, Xinchang, China). Jushi Group Co., Ltd. (Tongxiang, China) provided the fiberglass fabrics. The specific parameters of the glass fibre fabric are shown in Table 1. The polyimide film with a thickness of 0.1 mm was provided by Hangzhou Mick Chemical Instrument Co., Ltd. (Hangzhou, China) Silane coupling agent KH560 (purity ≥ 98.0%) was obtained from Qiangsheng Chemical Co., Ltd. (Suzhou, China) Glacial acetic acid (purity ≥ 99.5%) was sourced from Xinghua Chemical Co., Ltd. (Xinghua, China). The Xilong Science Co., Ltd. (Shantou, China) provided acetone (purity ≥ 99.8%).

### 2.2. Silane Coupling Agent Treatment

The glass fibre has poor holding properties, so a specific dose of infiltrant is added to the surface of the glass fibre before leaving the factory to facilitate subsequent weaving and reduce defects. However, the infiltrant will affect the bonding with polyphenylene sulphide resin to some extent and affect the mechanical properties of the composite. Soak the fiberglass fabric in acetone solution for 24 h, as shown in Figure 1b, remove the surface wetting agent and impurities, repeatedly rinse with distilled water, wash the residual liquid, and dry it in a vacuum-drying oven.

The dosage of silane coupling agent KH560 is an essential factor that directly affects the treatment effect. According to the guidance of silane coupling agent KH560 manufacturers and previous exploratory experiments, the dried glass fibre fabric was soaked in 2% silane coupling agent KH560, as shown in Figure 1c. Silane coupling agent KH560 was hydrolyzed fully under acidic conditions to better participate in the reaction, and glacial acetic acid was used to adjust the pH to 5 [40]. Finally, the glass fibre fabric was dried and was ready for use.

### 2.3. Preparation of Composite Materials

The DSC temperature scanning pattern of polyphenylene sulphide is shown in Figure 2.

Polyphenylene sulphide was scanned by DSC, and the heating rate was 20 °C/min. It can be seen from Figure 1 that the DSC curve reached its peak at 292.41 °C and its melting point was determined to be about 292.41 °C. According to Darcy’s Law [41,42], the impregnation rate V of resin to fibre satisfies the following relationship.
(1)$$V=\frac{\mathrm{dz}}{\mathrm{dt}}=-\frac{s}{\eta }\times \frac{\mathrm{dp}}{\mathrm{dz}}$$

In the formula, z is the impregnation depth, t is the contact time, s is the permeability, p is the pressure, and η is the matrix viscosity. The integral on both sides is
(2)$$\int {\mathrm{dz}}^{2}=\int \left(-\frac{s}{\eta }\mathrm{dp}\right)\mathrm{dt}$$
(3)$$z^{2}=-\frac{-2\mathrm{nd}}{\eta }\mathrm{dp}$$

The permeability of resin to fibre satisfies the following relationship,
(4)$$s=\frac{r^{2}}{4k}\times \frac{{\left(1-v\right)}^{3}}{v}$$

In the formula, s is the permeability, r is the fibre radius, v is the fibre volume content, and k is the Kozeny constant. The mass ratio, which is easy to control, is often used in experimental operations. The volume content and mass ratio satisfy the following relationship,
(5)$$v=\frac{m_{f}}{m_{f}+\frac{\rho_{f}}{\rho_{m}}{\;*\;m}_{m}}$$

In the formula, v is the volume content of fibre, $m_{f},\;m_{m}$ is the mass percentage of fibre and resin, $\rho_{f},{\;\rho }_{m}$ is the density of fibre and resin, respectively. The thickness of the fiberglass fabric used in this experiment is 0.25 mm (4 layers are 1 mm), the radius r is 2 μm, the mass percentage is 40%, and the density is 2.2 g/cm^3^. The minimum viscosity of polyphenylene sulphide is 200 Pa·s, and the density is 1.36 g/cm^3^. The Kozeny constant is 18. When the hot pressing pressure is 3 Mpa, it takes about 10 min to infiltrate the fabric completely. The hot pressing pressure was set to 3 Mpa and the hot pressing time was 10–15 min. The following experiments were set up (Table 2).

The composite layer is shown in Figure 3. The concrete layer is shown in Figure 4; the bottom layer 2 is polyimide film, the middle layer preform is polyphenylene sulphide resin and glass fibre fabric, and the top layer 4 is a polyimide film.

### 2.4. Characterization

#### 2.4.1. Surface Morphology

The longitudinal morphology of glass fibre treated with silane coupling agent KH560 and the bonding of the composites were observed by a scanning electron microscope (JSM-5610LV, Japan Co., Ltd., Tokyo, Japan). For better observation, the observed object needs to have good electrical conductivity, so the sample is gilded before the test.

#### 2.4.2. Infrared Spectrum Test

The glass fibre sample was dried in a vacuum oven at 80 °C for 12 h. The changes in glass fibre groups were observed by a Fourier transform infrared spectrometer (American thermoelectric company, model Nicolet 5700, Chicago, IL, USA), and the difference of glass fibre before and after modification was analyzed.

#### 2.4.3. Color Test

Under the condition of diffuse reflection, the condition of light source D65, and a 10° field of view, the color of each glass fabric reinforced polyphenylene sulphide composite surface was extracted by the DIGIEYE digital imaging color measuring system (VeriVide (GB)). Six different positions of color were selected for each sample, and the color value of CIE1976LAB was measured and averaged. The color value of the composite surface was based on the moulding temperature of 300 °C. The CIE1976LAB color difference formula is used to calculate the color difference of different composite surfaces.
(6)$${\Delta E}_{\mathrm{CIE}1976\left(L*a*b*\right)}={\left[{\left(\Delta L*\right)}^{2}+{\left(\Delta a*\right)}^{2}+{\left(\Delta b*\right)}^{2}\right]}^{\frac{1}{2}}$$

In the formula, ΔE represents the color difference, ΔL∗ represents the change value of light and dark, Δa∗ represents the change value of red and green, and Δb∗ represents the change value of yellow and blue.

#### 2.4.4. Assessment of Tensile Property

According to GB/T3354-2014, “Test method for tensile Properties of oriented fibre-reinforced Polymer Matrix Composites”, the tensile properties were tested by an MTS universal strength tester (Landmark, MTS system Shanghai Co., Ltd., Shanghai, China). The tensile strength of the sample is calculated according to the following formula.
(7)$$\sigma_{\tau }=\frac{P_{\max}}{\mathrm{ab}}$$
where $\sigma_{\tau }$ is the tensile strength, MPa. $P_{\max}$ is the maximum load, N. a is the sample width, mm. b is the sample thickness, mm.

#### 2.4.5. Assessment of Bending Strength 

According to GB/T3356-2014, “Test method for bending Properties of directional fibre-reinforced Polymer Matrix Composites”, the bending properties of samples were tested by a three-point bending test method using an MTS universal strength tester (Landmark, MTS system, Shanghai Co., Ltd.). The bending strength of the sample is calculated according to the following formula.
(8)$$\sigma_{f}=\frac{3P_{\max}\cdot L}{2\omega \cdot h^{2}}$$
where $\sigma_{f}$ is the bending strength, MPa. $P_{\max}$ is the maximum load of the specimen, N. L is span, mm. ω is the sample width, mm. h is the sample thickness, mm.

## 3. Results and Discussion

### 3.1. Effect of Modification of Silane Coupling Agent on Properties

#### 3.1.1. Surface Morphology before and after Silane Coupling Agent Treatment

The surface morphology of glass fibre before and after silane coupling agent treatment is shown in Figure 5.

Under a scanning electron microscope, the glass fibre sample in Figure 5a was treated with an acetone solution, and the surface of the sample was smooth and almost free of impurities. The glass fibre sample treated with silane coupling agent KH560 shows particle adhesion and protuberance on the surface of Figure 5b. Compared with Figure 5a, it can be seen in Figure 6b that a dense layer of silane coupling agent covers the sample’s surface.

The mechanism of glass fibre silane coupling agent KH560 treatment is shown in Figure 6. In the acidic environment with pH 5 regulated by glacial acetic acid, silane coupling agent KH560 is hydrolyzed, and the alkoxy groups in the molecular bond are hydrolyzed and condensed to form oligomeric siloxane containing Si-OH [43]. Oligomeric siloxane can form hydrogen bonds with the hydroxyl groups on the surface of glass fibre and can form covalent bonds with a water loss of glass fibre after drying. The principle of the experiment is shown in Figure 6.

#### 3.1.2. Infrared Spectrum Analysis

Figure 7 shows the infrared spectrum of the glass fibre before and after treatment with silane coupling agent KH-560. The antisymmetric stretching vibration of Si-O-Si in glass fibre is at 980 cm^−1^ wavenumbers, while the symmetrical stretching vibration of Si-O-Si in glass fibre is at 765 cm^−1^ wavenumbers. After treatment with silane coupling agent KH560, the absorption peaks at both sites become more robust due to the chemical reaction between the glass fibre surface and the silane coupling agent KH560. Combined with Figure 6, organosiloxane is hydrolyzed to form silanol, which then reacts with glass fibre to form a stable Si-O-Si bond structure.

#### 3.1.3. Micromorphology of Composites before and after Silane Coupling Agent Treatment

Figure 8 shows the SEM cross-section of the composite before and after silane coupling agent treatment.

The treatment of silane coupling agent KH560 can improve the bonding effect of glass fibre and polyphenylene sulphide. Figure 8a shows that before treatment with silane coupling agent KH560, the polyphenylene sulphide could not infiltrate the glass fibre sufficiently, and obvious voids could be seen at the junction of the glass fibre and polyphenylene sulphide. After modification by silane coupling agent KH560, shown in Figure 8b, the bonding effect of the glass fibre and polyphenylene sulphide have been improved, and there is no obvious gap between the glass fibre and the polyphenylene sulphide. On the fracture surface, it can be seen that the polyphenylene sulphide resin is still partially bonded to the glass fibre, and there is a good bonding property between the resin and the fibre. This is because silane coupling agent KH560 forms monolayers or two or three layers of molecular layers on the surface of the glass fibre by a hydrolysis reaction and combines glass fibre with polyphenylene sulphide by chemical bond [44,45]. At the same time, because of physical adsorption, the surface roughness and specific surface area of glass fibre with a smooth surface increased, and polyphenylene sulphide was easier to combine with glass fibre.

### 3.2. Effect of Moulding Temperature on Properties of Composites

#### 3.2.1. Effect of Forming Temperature on Mechanical Properties of Composites

The results of mechanical properties in Figure 9 show that the mechanical properties of glass fabric-reinforced polyphenylene sulphide composites increase at first and then decrease with the increase in moulding temperature. When the temperature is low, the fluidity of polyphenylene sulphide is relatively poor, so the bonding between polyphenylene sulphide and glass fibre fabric is uneven. There are many defects, resulting in stress concentration and a reduction in the properties of the composites. With the increase in temperature, the fluidity of the polyphenylene sulphide matrix increases, which can better infiltrate the glass fibre fabric. Good wettability can increase the contact area between the matrix and fibre, which is beneficial to improving the mechanical properties of composites. When the moulding temperature is too high, the composites’ mechanical properties decrease because the bivalent sulfur atoms in the polyphenylene sulphide molecular chain are unstable and easily oxidized at high temperatures, resulting in the decline of the overall mechanical properties [46]. When the temperature is 310 °C, the mechanical properties of glass fabric-reinforced polyphenylene sulphide composites are the best, and the tensile strength and bending strength are 51.9 MPa and 78 MPa, respectively.

Figure 10 shows the principle of the polyphenylene sulphide oxidation reaction. When the temperature is low, polyphenylene sulphide will undergo chain amplification (a), crosslinking oxidative response (b), and thermal crosslinking reaction (c), which can improve its relative molecular weight to a certain extent. In contrast, the tensile and bending properties of the composites can be improved. When the temperature increases, the crosslinking reaction of polyphenylene sulphide will still occur, but oxidative degradation plays a major role. The high temperature will cause the breaking of the C-S bond in the molecular chain of polyphenylene sulphide. Oxygen in the air will further destroy the formation of sulfur dioxide, carbon dioxide, and other gases that are released, while the formation of diphenyl ether and other compounds [47,48]. The oxidative degradation mechanism of polyphenylene sulphide is shown in Figure 11d–f.

#### 3.2.2. Effect of Forming Temperature on the Surface Color of Composites

Figure 11 shows the surface of glass fabric-reinforced polyphenylene sulphide composites at different moulding temperatures.

The moulding temperature will directly affect the properties of the polyphenylene sulphide. If the temperature is too high, the divalent sulfur atoms in the molecular chain of polyphenylene sulphide will be oxidized, and the mechanical properties will decrease. As seen from Figure 11, with the gradual increase in temperature, the surface colour of glass fibre-reinforced polyphenylene sulphide composites gradually deepens, and the overall colour tends to darken. Through the extraction of the surface color of composite materials at different moulding temperatures, the excellent difference ΔE is calculated based on the color value of the composite surface color at 300 °C. It can be seen from Table 3 that the value of ΔE increases with the increase in moulding temperature. It also shows that the color difference of the composite surface is increasing, among which ΔL* changes the most. Combined with Figure 11, it can be seen that the colour of the composite surface darkens gradually with the increase of moulding temperature, so the ΔE value is used to characterize the oxidation degree of polyphenylene sulphide.

## 4. Conclusions

This paper investigates the processing of polyphenylene sulphide composites. The polyphenylene sulphide composites are prepared by laminated hot pressing using glass fibre fabric as reinforcement and polyphenylene sulphide as the matrix. After the glass fibres are treated with silane coupling agent KH560, the initially smooth surface of the glass fibres forms bumps and the silane coupling agent KH560 forms a dense layer on the surface of the glass fibres. In glass fibre fabric reinforced polyphenylene sulfide composites, the interface between fibre and resin is good, and the bond between fibre and matrix is close. There are no longer any apparent voids. As the moulding temperature increases, the bending and tensile strengths of glass fibre fabric reinforced polyphenylene sulphide composites show a trend of increasing and then decreasing. The surface color of the glass fibre reinforced polyphenylene sulphide composites gradually deepened, and the overall color tended to become darker. When the hot pressing pressure was 3 Mpa, the hot pressing time was 10–15 min, and the hot pressing temperature was 300 °C, the mechanical properties of glass fibre fabric reinforced polyphenylene sulphide composites reached the optimum, and the tensile strength and bending strength were 51.9 MPa and 78 MPa, respectively.

## Figures and Tables

**Figure 1 materials-15-09036-f001:**
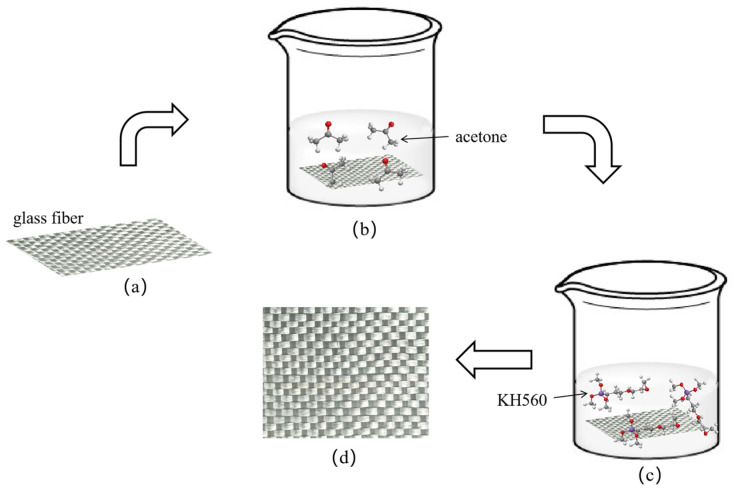
Reaction process. (**a**) Glass fiber before modification (**b**) Acetone treatment (**c**) Silane coupling agent treatment (**d**) Modified glass fiber.

**Figure 2 materials-15-09036-f002:**
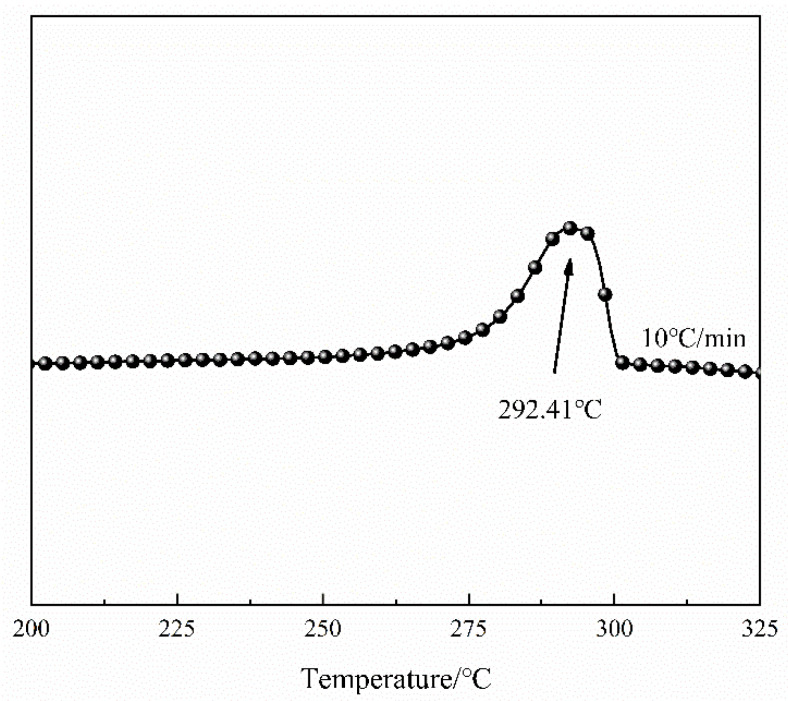
DSC heating scan pattern of polyphenylene sulphide.

**Figure 3 materials-15-09036-f003:**
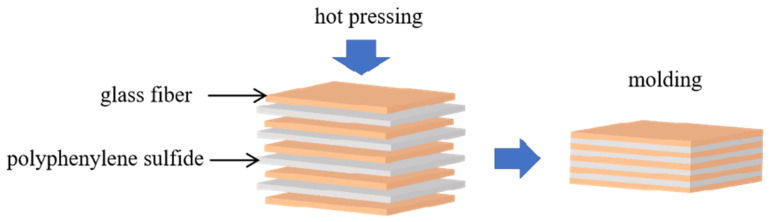
Prefab schematic.

**Figure 4 materials-15-09036-f004:**
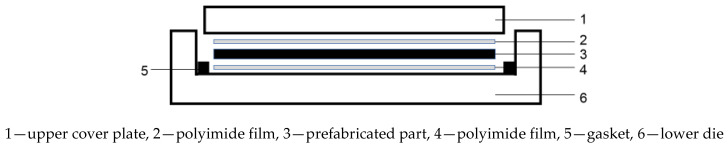
Schematic diagram for the arrangement of the hot pressing.

**Figure 5 materials-15-09036-f005:**
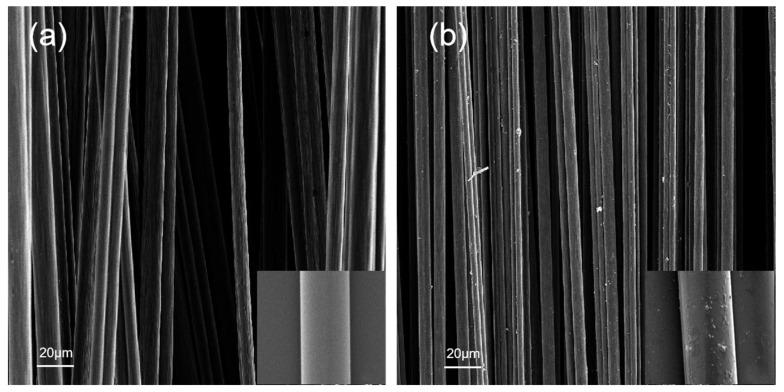
Microscopic morphology of the glass fibre surface (**a**) glass fibre before modification (**b**) glass fibre after modification.

**Figure 6 materials-15-09036-f006:**
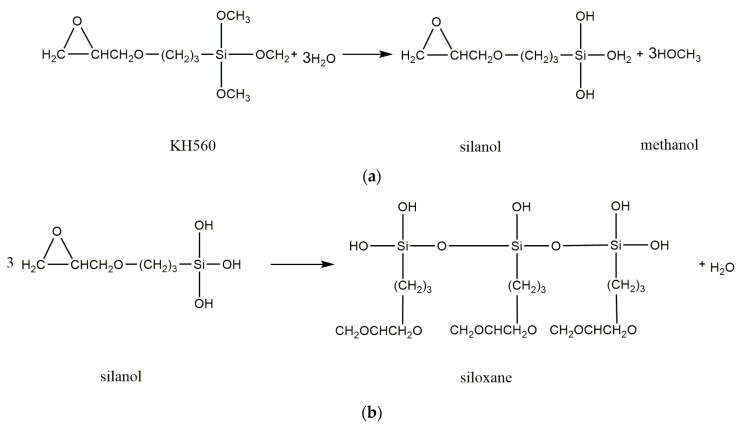
Treatment principle of silane coupling agent KH560. (**a**) Silane coupling agent hydrolysis. (**b**) Polycondensation to form oligomeric siloxane. (**c**) Oligosiloxane and glass fabric lose water to form a covalent bond.

**Figure 7 materials-15-09036-f007:**
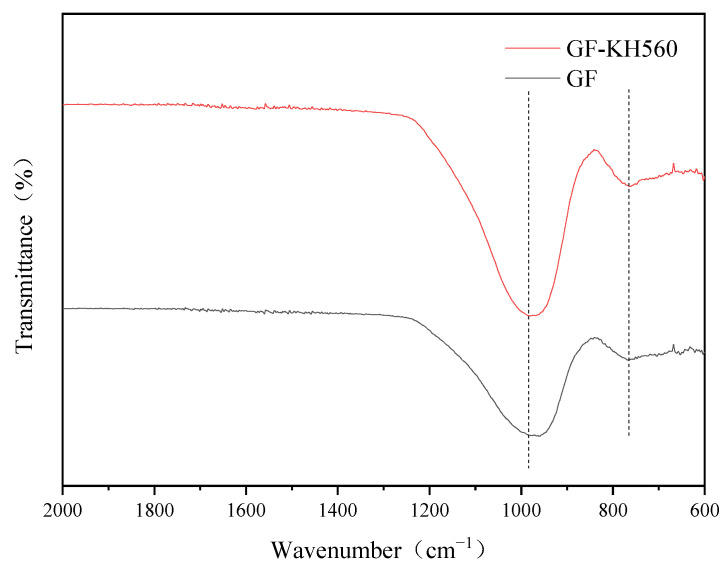
Infrared spectrum of glass fibre before and after modification.

**Figure 8 materials-15-09036-f008:**
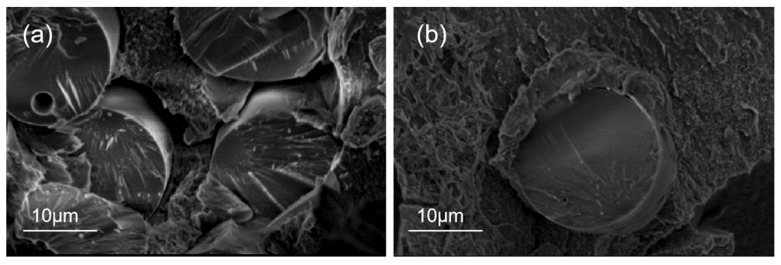
SEM cross-sectional photos of composites before and after treatment with silane coupling agent (**a**) pre-modified SEM (**b**) modified SEM.

**Figure 9 materials-15-09036-f009:**
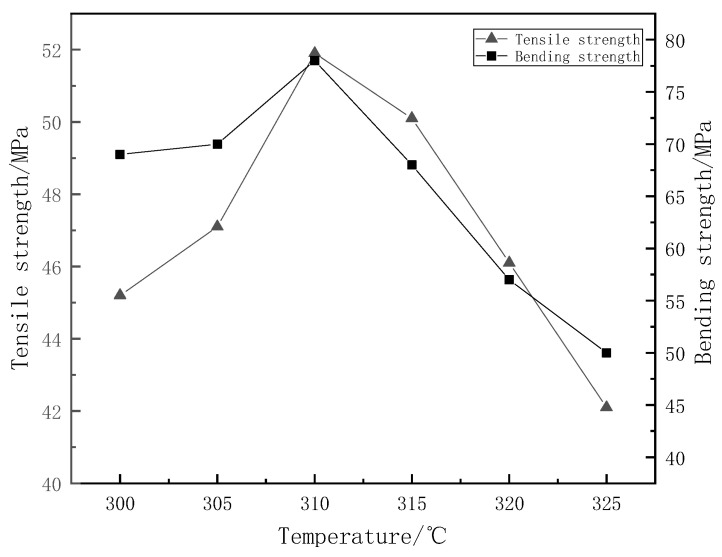
Effect of different moulding temperatures on the mechanical properties of glass fibre reinforced polyphenylene sulphide composites.

**Figure 10 materials-15-09036-f010:**
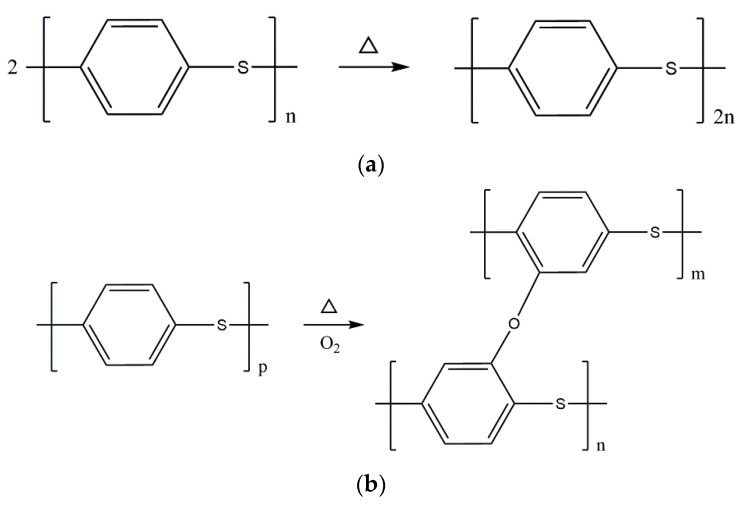
The principle of the oxidation reaction of polyphenylene sulphide. (**a**) chain amplification. (**b**) oxidative crosslinking reaction. (**c**) Thermal crosslinking reaction. (**d**) C Mel S bond breakage. (**e**) oxidative damage. (**f**) further oxidative damage.

**Figure 11 materials-15-09036-f011:**
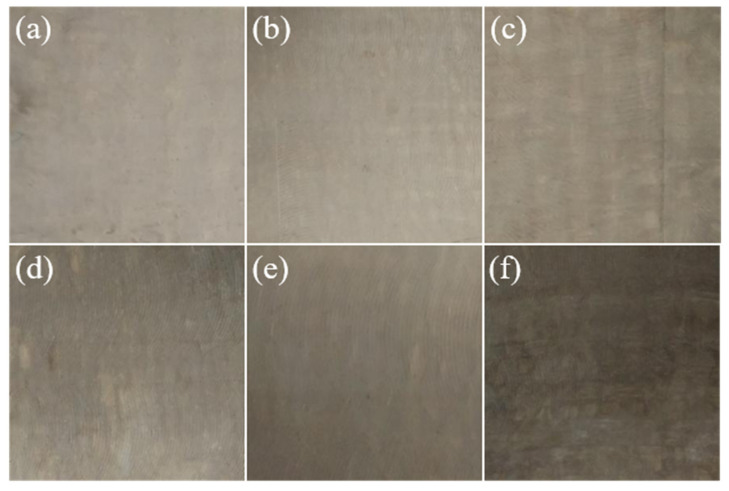
Surfaces of glass fibre reinforced polyphenylene sulphide composites at different moulding temperatures. (**a**) 300 °C (**b**) 305 °C (**c**) 310 °C (**d**) 315 °C (**e**) 320 °C (**f**) 325 °C.

**Table 1 materials-15-09036-t001:** Glass fibre fabric specifications.

Index	Specification
warp density	34 yarns/10 cm
weft density	25 yarns/10 cm
Warp yarn density	440 tex
Weft yarn density	380 tex
GSM	250 g/m²
Thickness	0.25 mm

**Table 2 materials-15-09036-t002:** Experimental parameters for composite preparation.

Sample Number	Variable	Silane Coupling Agent Modification	Temperature/°C
1	Silane coupling agent modification	No	310
2		Yes	310
3	Temperature	Yes	300
4		Yes	305
5		Yes	310
6		Yes	315
7		Yes	320
8		Yes	325

**Table 3 materials-15-09036-t003:** Surface colour extraction of glass fibre reinforced polyphenylene sulphide composites at different moulding temperatures.

Temperature/°C	L	a	b	ΔL	Δa	Δb	ΔE
300 °C	64	1	8	—	—	—	—
305 °C	63	1	10	−1	0	2	2.24
310 °C	59	2	11	−5	1	3	6.00
315 °C	53	1	9	−11	0	1	11.05
320 °C	48	1	10	−16	0	2	14.14
325 °C	35	2	9	−19	1	1	19.05

## Data Availability

Not applicable.

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
