# Peer review of "Study on Preparation and Properties of Glass Fibre Fabric Reinforced Polyphenylene Sulphide Composites"

_materials, 2022, doi:10.3390/ma15249036_

Round 1

Reviewer 1 Report

The authors presented their work on preparation and characterization of glass fiber fabric reinforced polyphenylene sulfide composites. The manuscript is written in the way that is hard to follow. A significant proofreading and rewriting should be done. Briefly, some of the issues to be also addressed:

- Abstract is too long and should be rewritten in more concise way.

- Introduction needs to be improved. Also, if the molecular structure of polyphenylene is discussed, it is better to include it.

- Line 108-110. Check the grammar.

- Why DSC data is located in paragraph 2.3 while the procedure for DSC is in paragraph 2.4.1? DSC data should be shifted to the Results Section.

- Check the equations.

- Line 148. What experimental results? Why that was mentioned in Experimental Section?

- Table 2. It is suggested to be modified. It is unclear what conditions belong to what variable.

 -2.4.3 Lines 181-187. This information should not be located in Experimental Section.

- Line 202. Tensile speed was tested...?

- Line 215-217. Check the formula and the description

- It is suggested to rewrite and proofread the entire section.

- Line 229. Compared to Figure 6(a)... What is compared?

-Figure 7. Element "Si" has to be capitalized.

- Figure 10. The figure has to be replaced with the one in a higher quality.

- Conclusions. It is suggested to make the section more concise. 

Reviewer 2 Report

Reviewing report for the Manuscript materials-2054012

In this study, the authors reported the synthesis glass fiber-reinforced polyphenylene sulfide composites by laminated hot pressing method to be used as reinforcement and polyphenylene sulfide as the matrix, glass fiber fabric was treated with silane coupling agent KH560, and composites were prepared by laminated hot pressing method. The effect of process parameters on the properties of glass fiber fabric-reinforced polyphenylene sulfide composites was studied.

Generally, the findings of this study are interesting, and the data are clearly discussed, however, there are some revisions should be made for this manuscript to accepted for publication in “materials”. The suggested revisions are noted below:

 1.      The abstract of the manuscript needs to be revised and re-presented.

2.      Figure 1 didn’t have any information; the authors need to add the details such as name of the components on the figure.  

3.      The authors didn’t reefer to Figure 1 in the main text, please cite Figure 1 in the main manuscript.

4.      In Figure 7, the authors recommended to add the name of the reactants and the products in the chemical reaction equations.

5.      Why didn’t the authors provide FT-IR of the glass fiber before and after silane coupling agent treatment to show the chemical bonding?

6.      The conclusion section is too large, needs to be smaller and concise. The authors repeat sentences from lines (362-371) "Good wettability can increase the contact area between matrix and fiber, which is beneficial to the improvement of the mechanical properties of the composites. When the molding temperature is 310-325 ℃, the bivalent sulfur atoms in the molecular chain of polyphenylene sulfide resin are unstable and easily oxidized at high temperatures, the overall color of the composites becomes darker and darker, and the overall mechanical properties of the composites decreased. When the hot-pressing pressure is 3 MPa, the hot-pressing time is 10-15min, and the hot pressing temperature is 310 ℃, the mechanical properties of glass fabric-reinforced polyphenylene sulfide composites reach the best, the tensile strength reaches 51.9 MPa, and the bending strength reaches 78 MPa " ……..This is the same in line 289-296.  

Reviewer 3 Report

Reviewer’s report:

Title: Study on preparation and properties of glass fiber fabric reinforced polyphenylene sulfide composites.

In this work, authors reported composite fabrication with glass fiber fabric as reinforcement and polyphenylene sulfide as matrix with silane coupling agent KH560 for glass fiber fabric treatment and laminated hot pressing method is used to fabricate the composite. The manuscript has enough data with a good discussion which may beneficially for other researchers. However, it still needs revision and improvement in some parts as shown in the comments below.

1.  At Fig.1, add labels for each step and then give some detail information in the figure caption for those steps.

2. At Fig.3, add information what is the laminated layer which is represented with 2 colors at the schematic of the sandwich structure.

3. At Fig. 5, what means of Ysv, YLv, and YSL line directions, you may can add some information in the figure caption.

4. Don’t make a single paragraph which consist only a very few lines (e.g., lines 140-141, 142-143, 144-145, 165-167, and other). A good paragraph normally consists of around 10 lines.

5. Explanation of Equation 8 at lines 216-217 are not accurate. “1” supposes to be σf ?, “H” supposes to be “h”?, what is Pmax?

6. At line 254, how many degrees for the smaller contact angle? Add the result of the contact angle measurement test.

7. At line 282, typo: Figure 8 is a contact angle test, it should be Fig. 10.

8. At Fig.11(c), an information is still in Chinese character, you need to make it in English.

9.  To verify the water compatible functional groups which related to the wettability improvement, it will beneficially if it can be added further surface evaluation like FTIR measurement test.

Reviewer 4 Report

·       Discuss more about practical applications compared with the literature.

·       Add more information in introduction about composites applications.

·       In the Introduction section, the authors cited the specific results of previous research and cited them adequately. However, they did not mention their shortcomings in previous research. In the Introduction section, the penultimate paragraph should contain common features of previous research. The shortcomings of previous research should also be pointed out, in general.

·       In the Introduction section, the last paragraph should contain the scientific contribution and scientific hypotheses of your research. Complete, further elaborate the scientific contribution and scientific hypotheses of your research. Be explicit. In addition to the goal of the research (which was written), the novelty in the context of the scientific contribution should be pointed out. Scientific contributions should be written based on the shortcomings of previous research in the literature. In this way, the authors will better emphasize novelty and scientific soundness.

·       The study also needs to have a discussion where the authors need to discuss their findings and compare them with the previous studies, either review or empirical ones.

·       The discussion should at least conclude something understandable.

·       In the conclusions, state the scientific contribution, the shortcomings of your methodology and future research.

·       The article must be put in the format.

Round 2

Reviewer 4 Report

Paper was improved.